# The Validity of Wireless Earbud-Type Wearable Sensors for Head Angle Estimation and the Relationships of Head with Trunk, Pelvis, Hip, and Knee during Workouts

**DOI:** 10.3390/s22020597

**Published:** 2022-01-13

**Authors:** Ae-Ryeong Kim, Ju-Hyun Park, Si-Hyun Kim, Kwang Bok Kim, Kyue-Nam Park

**Affiliations:** 1Department of Rehabilitation Science, Jeonju University, Jeonju 55069, Korea; pjstar3@jj.ac.kr (A.-R.K.); juju1541@jj.ac.kr (J.-H.P.); 2Department of Physical Therapy, Sangji University, Wonju 26339, Korea; sihyunkim@sangji.ac.kr; 3Digital Health Care R&D Department, Korea Institute of Industrial Technology, Cheonan 31056, Korea; kb815kim@kitech.re.kr; 4Department of Physical Therapy, Jeonju University, Jeonju 55069, Korea

**Keywords:** head, motion, wearable sensor, wireless earbud, workout, validity

## Abstract

The present study was performed to investigate the validity of a wireless earbud-type inertial measurement unit (Ear-IMU) sensor used to estimate head angle during four workouts. In addition, relationships between head angle obtained from the Ear-IMU sensor and the angles of other joints determined with a 3D motion analysis system were investigated. The study population consisted of 20 active volunteers. The Ear-IMU sensor measured the head angle, while a 3D motion analysis system simultaneously measured the angles of the head, trunk, pelvis, hips, and knees during workouts. Comparison with the head angle measured using the 3D motion analysis system indicated that the validity of the Ear-IMU sensor was very strong or moderate in the sagittal and frontal planes. In addition, the trunk angle in the frontal plane showed a fair correlation with the head angle determined with the Ear-IMU sensor during a single-leg squat, reverse lunge, and standing hip abduction; the correlation was poor in the sagittal plane. Our results indicated that the Ear-IMU sensor can be used to directly estimate head motion and indirectly estimate trunk motion.

## 1. Introduction

Athletes and active people have continued exercising during the coronavirus disease 2019 (COVID-19) pandemic quarantine period to maintain their physical fitness, decrease pain and fatigue, and prevent injury [1]. The results of a survey completed by 606 respondents from 132 countries indicated that squats, single-leg squats, push-ups, and lunges were the most commonly applied workouts during by athletes the COVID-19 quarantine period [1]. The COVID-19 pandemic has led to rapid changes in the traditional workout industry (from contact to noncontact workout settings) via online platforms and wearable devices [2]. Wearable sensors have the advantages of portability, suitability for indoor workouts, facilitation of large population analysis, and longitudinal tracking of motion data for patients with musculoskeletal pain [3]. Wearable waist-belt and wristband sensors can be used to continuously monitor pain intensity in patients with chronic back pain and knee pain, and for investigating the relationship between wearable sensor data and pain intensity [3]. Wearable sensors can also be used for managing chronic musculoskeletal pain in digital therapy programs, including sensor-guided exercise therapy and behavioural health coaching [4], such that they helped with reducing pain intensity and improving exercise adherence in a longitudinal study of 10,000 patients with lower back and knee pain [3].

Inertial measurement units (IMUs) are commonly used in wearable devices to capture and monitor human motion [5,6]. The validity of IMU sensors was demonstrated when measuring the acceleration or joint angle during squats, single-leg squats, walking, and running exercises, compared with a 3D motion analysis system [7,8,9]. According to a review article, three or more IMU sensors were used on the chest, lumbar region, upper limbs, and lower limbs for motion analysis in 37 previous studies [5]. The accuracy of kinematic measurement increases in proportion to the number of IMU sensors attached to the body [10]. However, the attachment of multiple sensors negates the wearable sensor’s advantages of portability, comfort, and ease of acceptance by users in real-world environments [11]. The smartwatch, which has only a single IMU sensor, is the most popular wearable device for the assessment of physical activity via energy expenditure and steps during daily activities [10,12]. However, because the smartwatch relies on the degrees of upper limb motion [6], the data are subject to motion artefacts and do not measure the trunk motion during squat and lunge exercises; notably, trunk kinematic data in the sagittal or frontal plane have been used for effective training (e.g., to contract hip and knee muscles) and screening (e.g., to identify people developing knee pain during lunge and squat exercises) [11,13]. Excessive, uncontrolled trunk motion is also seen in people with knee pain and chronic ankle instability during single-leg squats [14,15], so the assessment of trunk motion using wearable sensors attached to the body would be valuable in sports rehabilitation.

In a study in which measurements were taken using IMU sensors placed inside a baseball cap and attached to the waist of healthy subjects while walking, the head motion was strongly correlated with trunk motion based on acceleration data [16]. When visual feedback to young or older healthy adults on trunk translation in the anteroposterior and mediolateral positions while walking was provided, the head–neck translation decreased significantly, implying that head motion is related to trunk motion [17]. Head motion data can help minimise horizontal head translation while walking in older people and patients with chronic diseases such as Parkinson’s disease, multiple sclerosis, and vestibular impairment [16,18]. To measure head motion, the current study used wireless earbuds, which are increasingly popular wearable devices [19]. A wireless earbud brings the sensor closer to the trunk than a wrist-worn smartwatch would; therefore, an earbud IMU sensor can potentially estimate head and trunk motion more accurately during workouts.

Wearable sensors have been used to prevent injury in healthy athletes, which is an important element of modern healthcare [20]. Incorrect motion sequences lead to musculoskeletal pain after exercise, halting exercise training [21]; for instance, excessive trunk lateral lean during single-leg squats and an inappropriate amount of trunk forward bending during squats can induce knee and lower back pain after training [15,22]. Excessive lateral trunk leaning during single-leg drop jumps in healthy athletes was found to predict future non-contact knee injury in a 1-year prospective study [23]. Therefore, a study of the validity of wireless earbud IMU sensors for measuring head and trunk motions is key for assessing whether wearable sensors can predict and prevent injuries in healthy active people. Thus, the primary purpose of the present study was to confirm the concurrent validity of wireless earbud-type IMU sensors for estimating head angle, compared with a 3D motion analysis system during workout activities (squat, single-leg squat, lunge, and standing hip abduction) in healthy active adults. The secondary purpose was to investigate the pairwise relationships between head angle (measured by each IMU sensor and the 3D motion analysis system) and the angles of other joints (trunk, pelvis, hip, and knee, measured by the 3D motion analysis system) to determine which joints show movement patterns similar to head movement during workout activities.

## 2. Materials and Methods

### 2.1. Participants

The study population consisted of 20 healthy volunteers from XX University, all of whom regularly engaged in exercise training or workouts (Table 1). The inclusion criteria were age = 20 years; 1–3 h of regular workout activities per day, 1–3 days per week; ability to fully engage in squats, single-leg squats, lunges, and standing hip abduction activities without pain in the lower back or lower extremities for at least 1 month prior to the study. The exclusion criteria were prior lower back or lower extremity surgery, current pregnancy, or existing neurological impairments in cognitive or vestibular systems [24,25]. This study was conducted in accordance with the ethical principles for medical research involving human participants, established by the Declaration of Helsinki. The study was approved by the Institutional Review Board of Jeonju university (approval No: jjIRB-210114-HR-2021-0113) and all participants provided written informed consent prior to enrolment.

### 2.2. Instruments

The experimental study was carried out in a laboratory setting. While participants performed four exercises, kinematic data in sagittal and frontal planes were recorded simultaneously using both the earbud-type IMU sensor and a 3D motion analysis system (Vicon, Oxford, UK). The 3D motion analysis system was used as a reference to collect kinematic data via six infrared cameras at a frequency of 100 Hz. Signals emitted from the Vicon camera were reflected from markers attached to anatomical landmarks on the participant according to the full-body plug-in-gait marker model. In this study, 39 markers measuring 15 mm^2^ each were placed on each participant: on the second metatarsal head, posterior calcaneus and fibular lateral malleolus of both feet, both shins, lateral epicondyles of both knees, both thighs, anterior and posterior superior iliac spine, 7th cervical and 10th thoracic vertebrae, sternum, clavicle, right scapula, superior to both acromioclavicular joints, both upper arms, both lateral epicondyles of the elbow, both forearms, medial and lateral styloid processes of both wrists, and 2nd metacarpals bilaterally; four other markers were attached to a headband [26].

A single IMU (36 × 15 × 7.5 mm; weight including battery: 8 g) was embedded into the right wireless earbud (QCY-T6, Dongguan Hele Electronics Co., Ltd., Dongguan, China) (Figure 1). To measure linear acceleration and angular rotation, a high-resolution IMU (BNO080; Ceva Technologies, Rockville, MD, USA) was equipped with a triaxial accelerometer and triaxial gyroscope. The orientations of the axes of the accelerometer were aligned with the gravitational axis while the participant was standing. The accelerometer of the IMU recorded gravitational acceleration (g) in the vertical axis when the participant was in a standing position. IMU data which were collected by the Microprocessor (STM32G030, STMicroelectronics, Geneva, Switzerland) were sent through a BLE module with a sampling frequency of 100 Hz. Every sample contains signed 16 bits acceleration outputs for the *x*-, *y*- and *z*-axis. Acceleration outputs were scaled to be a proportional value of Earth gravity which are $G_{px}$, $G_{py}$ and $G_{pz}$. Full-scale IMU sensor value (=8 G) was equal to the maximum value of singed 16 bits integer.

The orientation of the IMU can be defined with its roll(ϕ), pitch(θ), and yaw(ψ). The roll, pitch, and yaw matrices are as follows:(1)$$R_{x}\left(\phi \right)=\left(\begin{array}{ccc} 1 & 0 & 0\\ 0 & \cos\phi & \sin\phi \\ 0 & -\sin\phi & \cos\phi \end{array}\right)$$
(2)$$R_{y}\left(\theta \right)=\left(\begin{array}{ccc} \cos\theta & 0 & -\sin\theta \\ 0 & 1 & 0\\ \sin\theta & 0 & \cos\theta \end{array}\right)$$
(3)$$R_{z}\left(\psi \right)=\left(\begin{array}{ccc} \cos\psi & \sin\psi & 0\\ -\sin\psi & \cos\psi & 0\\ 0 & 0 & 1 \end{array}\right)$$

When the IMU sensor is rotated, Earth’s gravitation field vector g is rotated by pitch, roll, and yaw matrices
(4)$$R_{y}\left(\phi \right)R_{x}\left(\theta \right)R_{z}\left(\psi \right)\left(\begin{array}{c} \begin{array}{c} 0\\ 0 \end{array}\\ 1 \end{array}\right)=\left(\begin{array}{c} \begin{array}{c} -\sin\theta \cos\phi \\ \sin\phi \end{array}\\ \cos\theta \cos\phi \end{array}\right)$$

When arbitrary accelerometer reading is $G_{p}$,
(5)$$\frac{1}{\sqrt{G_{px}{}^{2}+G_{py}{}^{2}+G_{pz}{}^{2}}}\left(\begin{array}{c} \begin{array}{c} G_{px}\\ G_{py} \end{array}\\ G_{pz} \end{array}\right)=\left(\begin{array}{c} \begin{array}{c} -\sin\theta \cos\phi \\ \sin\phi \end{array}\\ \cos\theta \cos\phi \end{array}\right)$$

Solving for the roll and pitch angles gives
(6)$$\phi =\tan^{-1}\left(\frac{G_{py}}{\sqrt{G_{px}{}^{2}+G_{pz}{}^{2}}}\right)$$
(7)$$\theta =\tan^{-1}\left(\frac{-G_{px}}{G_{pz}}\right)$$

Before angle calculation, the accelerometer output was filtered with a low-pass filter to remove linear acceleration caused by the participant’s motion. Without this removal of linear acceleration, the acceleration added by the participant’s movement increased the error of the estimated angles. To properly calculate roll and pitch angle from an accelerometer, the magnitude of accelerometer output should be near 1 G, which means only gravitational acceleration exists. After low-pass filtering, the magnitude of accelerometer output converged to 1 G within a 5% difference.

For offset calibration, pre-measurement was performed for 1 s when measurement began. The measured mean offsets in sagittal and frontal angles were used to remove offsets. During a 1 s calibration period, a subject was told to remain stationary, 100 data samples were collected and assessed to confirm calibration. Only when (Maxϕ−Minϕ) and (Maxθ−Minθ) are less than 0.5°, it was accepted as a successful calibration. If not, a test started over from the calibration period.

### 2.3. Procedure

This experimental study consisted of three stages: general preparation, warm-up, and data acquisition.

General preparation stage:

All participants provided responses to questions regarding demographic characteristics, exercise frequency per week, and daily exercise duration (Table 1); they also confirmed that they had no surgical history and no current musculoskeletal pain or symptoms. Anthropometric data (leg length and knee, ankle, elbow, and wrist width, hand thickness, and shoulder offset) were measured with a measuring tape and digital calliper prior to use of the 3D motion analysis system [27,28].

2.Warm-up stage:

The participants performed 5 min of indoor cycling as a warm-up while wearing their own shoes; they then engaged in a 5 min rest period [29]. The wireless earbud-type IMU sensor was worn on the right ear. To record kinematic data using the 3D motion analysis system, 39 markers were attached using double-sided tape with the participant in the standing position. To become familiar with the procedure, each participant was asked to perform each workout activity three times while watching a video on a screen set at eye level directly in front of them.

3.Data acquisition stage:

The order of the four exercises (reverse lunge, squat, single-leg squat, and standing hip abduction) was randomised using Excel (Microsoft, Redmond, WA, USA). Each workout was performed three times at a self-selected speed with a 1 min break time between workouts. The supporting leg during a single-leg squat, reverse lunge, and standing hip abduction was regarded as the supporting side when kicking a soccer ball. To synchronise the two datasets obtained with the 3D motion analysis system and IMU sensor, participants were asked to perform rapid neck flexion at the beginning of each workout [30]. Each workout was performed while gazing at a screen placed at eye level directly in front of the participant, and the specific methods of each workout were explained as follows (Figure 2):

Squat: The participant standing with hands placed on the waist and the feet shoulder-width apart was asked to perform a mini-squat to a self-determined depth within an angle of 90° knee flexion, then return to the starting position [31].

Single-leg squat: The participant stood on the single supporting leg with the hands placed on the waist, then squatted down and up while maintaining balance. The squat depth was not controlled because this study focused on capturing the true movement patterns of the participants [9].

Reverse lunge: The reverse lunge was initiated by moving the non-supporting foot backwards, while the supporting foot remained planted. The front knee continued to flex until achieving a position parallel to the floor; the back knee was brought closer to the floor using a 6 cm foam pad as a tactile depth cue. The distance between the front foot and back foot was approximately 75% of leg length. The participant was then asked to return to the starting position [32].

Standing hip abduction: The participant was asked to abduct the arm of the non-supporting side to 90° while standing with the feet shoulder-width apart. Subsequently, the participant was asked to abduct the hip of the non-supporting leg as far as possible and then return to the starting position [33].

### 2.4. Data Processing

The raw data of head, trunk, pelvis, hip, and knee obtained from the 3D motion analysis system were analysed using Nexus software ver. 1.8.5 (Oxford Metrics Ltd., Oxford, UK); they were filtered using a low-pass, zero-lag, 4th order Butterworth filter with a cut-off frequency of 6 Hz [8]. The angle data in the sagittal and frontal planes were calculated based on global coordinates by deriving ‘YXZ’ Cardan angles. The raw IMU data were filtered using a MATLAB program (MathWorks). First, raw acceleration data were filtered to remove linear acceleration using a low-pass, 2nd order Butterworth filter with a cut-off frequency of 1 Hz. The sagittal and frontal plane angles were estimated using filtered acceleration data. To synchronise the two datasets, each peak of rapid neck flexion obtained by the 3D motion analysis system and IMU data was matched; the concurrent validity was analysed (Figure 2) [30].

### 2.5. Statistical Analysis

Statistical analysis was performed with SPSS (ver. 25.0; SPSS, Chicago, IL, USA). The alpha level for statistical significance was set at 0.05. According to the results of the Shapiro–Wilk test for the normality of the data distribution, the Spearman rho correlation coefficient (ρ) was used to investigate the concurrent validity of head angle data between the IMU sensor and the 3D motion analysis system; the pairwise correlations between head angle (determined with the IMU sensor) and the angles of the trunk, pelvis, hip, and knee (recorded with the 3D motion analysis system); the pairwise correlations between the head angle and the angles of the trunk, pelvis, hip, and knee (all recorded with the 3D motion analysis system). The correlations were expressed as perfect (ρ = 1), very strong (0.8 ≤ ρ < 1), moderate (0.6 ≤ ρ < 0.8), fair (0.3 ≤ ρ < 0.6), or poor (0 < ρ < 0.3) [34].

## 3. Results

### 3.1. Concurrent Validity of the Earbud Type IMU Sensor

The observed concurrent validities of the head angle determined with the earbud-type IMU sensor, compared with the angle determined with the 3D motion analysis system, were very strong during the squat, single-leg squat, and standing hip abduction exercises; the concurrent validity was moderate during reverse lunge in the sagittal plane. In addition, the head angle measured with the earbud-type IMU sensor showed moderate validity in the frontal plane, compared with the 3D motion analysis system, during all workout activities (Table 2).

### 3.2. Pairwise Correlations between Head Angle Determined with the IMU Sensor and Angles of Other Joints Determined with the 3D Motion Analysis System

We investigated the pairwise correlations between the head angle (determined with the IMU sensor) and the angles of other joints (determined with the 3D motion analysis system) to determine which joints had movement similar to head movement during the workout. In the frontal plane, the trunk angle showed a fair correlation with IMU sensor head angle during the single-leg squat, reverse lunge, and standing hip abduction. In addition, the pelvis and knee angles showed fair correlations with head angle during the single-leg squat and standing hip abduction exercises, respectively. The pairwise correlations between the head angle and the angles of other joints in the sagittal plane were poor (Table 3).

### 3.3. Pairwise Correlations between Head Angle and Angles of Other Joints Determined with the 3D Motion Analysis System

For comparison, we also investigated the pairwise correlations between head angle and the angles of other joints using only the 3D motion analysis system data. These correlations were similar to the results when using both the IMU sensor and the 3D motion analysis system. Specifically, the trunk angle in the frontal plane showed a moderate or fair correlation with the head angle during the single-leg squat, reverse lunge, and standing hip abduction exercises; the pelvis angle in the frontal plane showed a fair correlation with the head angle during the single-leg squat exercise. The pairwise correlations between the head angle and the angles of other joints in the sagittal plane were poor (Table 3).
sensors-22-00597-t003_Table 3Table 3Pairwise correlations between the head and other joints during workout activities.

Spearman’s ρ (95% CI) in Sagittal PlaneSpearman’s Ρ (95% CI) in Frontal PlaneIMU vs. VICONVICON vs. VICONIMU vs. VICONVICON vs. VICONSquatTrunk−0.028 (−0.22, 0.164)0.043 (−0.166, 0.252)0.268 * (0.094, 0.442)0.222 (0.059, 0.384)Pelvis−0.005 (−0.173, 0.162)0.067 (−0.113, 0.246)0.002 (−0.145, 0.198)−0.035 (−0.153, 0.083)Hip_Rt0.019 (−0.166, 0.203)0.076 (−0.121, 0.274)0.069 (−0.086, 0.224)0.143 (0.01, 0.276)Hip_Lt0.014 (−0.169, 0.198)0.07 * (−0.127, 0.267)0.071 (−0.079, 0.221)0.071 (−0.102, 0.244)Knee_Rt−0.063 * (−0.268, 0.141)−0.006 * (−0.228, 0.216)−0.024 (−0.195, 0.148)0.021 (−0.148, 0.191)Knee_Lt−0.023 * (−0.22, 0.173)0.048 * (−0.166, 0.262)0.003 (−0.153, 0.159)0.021 (−0.131, 0.174)Single-leg squatTrunk−0.216 (−0.339, 0.094)−0.137 (−0.277, 0.003)0.536 * (0.405, 0.667)0.606 * (0.498, 0.715)Pelvis−0.122 (−0.25, 0.005)−0.069 (−0.204, 0.066)0.366 * (0.231, 0.501)0.33 * (0.182, 0.479)Hip_Rt−0.146 (−0.295, 0.003)−0.115 (−0.273, 0.043)−0.272 (−0.414, −0.129)−0.197 (−0.352, −0.041)Hip_Lt−0.086 (−0.221, 0.049)−0.049 (−0.186, 0.089)0.267 (0.118, 0.416)0.260 (0.114, 0.406)Knee_Rt−0.151 (−0.295, −0.007)−0.145 (−0.298, 0.008)0.258 (0.098, 0.417)0.276 (0.12, 0.431)Knee_Lt−0.121 (−0.23, −0.012)−0.101 (−0.212, 0.01)0.132 (0.017, 0.247)0.195 (0.068, 0.322)Reverse lungeTrunk−0.129 (−0.239, −0.019)−0.168 (−0.325, 0.011)0.437 * (0.335, 0.54)0.459 * (0.329, 0.589)Pelvis−0.138 (−0.225, −0.051)−0.193 (−0.327, −0.06)−0.005 (−0.134, 0.123)−0.008 (−0.156, 0.141)Hip_Rt−0.292 (−0.384, −0.2)0.009 (−0.089, 0.107)−0.148 (−0.242, −0.053)−0.027 (−0.142, 0.087)Hip_Lt0.076 (−0.064, −0.217)−0.014 (−0.166, 0.137)0.033 (−0.1, 0.166)−0.008 (−0.157, 0.141)Knee_Rt0.026 (−0.106, −0.158)0.099 (−0.05, 0.249)0.231 (0.103, 0.36)0.191 (0.051, 0.33)Knee_Lt0.094 (−0.036, −0.223)0.053 (−0.097, 0.202)0.189 (0.072, 0.305)0.198 (0.074, 0.323)Standing hip abductionTrunk−0.158 (−0.323, −0.007)−0.152 (−0.328, 0.025)0.384 * (0.185, 0.582)0.467 * (0.27, 0.665)Pelvis−0.183 (−0.349, −0.017)−0.187 (−0.359, −0.015)0.254 * (0.05, 0.458)0.234 * (0.022, 0.447)Hip_Rt−0.171 (−0.348, −0.006)−0.201 (−0.382, −0.02)−0.124 (−0.365, 0.117)−0.074 (−0.319, 0.171)Hip_Lt−0.146 (−0.312, −0.02)−0.15 (−0.323, 0.024)−0.155 (−0.354, 0.044)−0.105 (−0.313, 0.104)Knee_Rt−0.239 (−0.357, −0.122)−0.237 (−0.354, −0.119)0.337 * (0.145, 0.529)0.296 * (0.112, 0.479)Knee_Lt−0.155 (−0.227, −0.038)−0.132 (−0.272, 0.008)−0.095 (−0.188, 0.173)0.025 (−0.173, 0.223)* *p*-values were significant (*p* < 0.05). Rt: Right; Lt: Left.


## 4. Discussion

This study confirmed the moderate-to-very-strong validity of measuring the head angle during four types of workouts using an earbud-type IMU sensor, compared with a 3D motion analysis system. The head angle determined with the IMU sensor in the frontal plane showed a fair correlation with trunk angle determined with the 3D motion analysis system during the single-leg squat, lunge, and standing hip abduction activities. The earbud type-IMU sensor could be useful for measuring the motion of the head and trunk, as an alternative to the use of a 3D motion analysis system for workout motion-focused coaching, for active people who prefer to listen to music during workouts.

### 4.1. Concurrent Validity of the Earbud-Type IMU Sensor

Comparison of head angles measured using the earbud-type IMU sensor and the 3D motion analysis system confirmed moderate-to-very-strong validity in the sagittal and frontal planes during the squat, single-leg squat, reverse lunge, and standing hip abduction exercises. The validity in the sagittal plane was higher than the validity in the frontal plane in this study. Previous studies suggested that the higher validity in the sagittal plane, compared with the frontal plane, was caused by greater angles in the sagittal plane during squats and single-leg squats, as determined using a Kinect camera and 3D motion capture system [9,35,36]. Although there have been no reports on the validity of head-worn IMU sensors for measuring kinematic data during workouts, some reports were consistent with our findings regarding functional activities and standing head motions [24]. A previous study comparing acceleration between an IMU sensor embedded into eyeglasses and a 3D motion analysis system reported weak-to-strong validity during sit-to-stand motion (Pearson r = 0.37–0.94 for the vertical axis and 0.36–0.61 for the anteroposterior axis) [24]. Two other studies also found good agreement between IMU sensors attached to the forehead and a 3D motion-capture system for measuring head angles during head flexion/extension, lateral bending, and rotation in standing healthy young subjects [37,38]. In individuals with and without mild traumatic brain injury, IMU sensors attached to the forehead exhibited moderate-to-excellent validity for estimating head angles while walking and turning or nodding the head [39]. The IMU sensor locations (eyeglasses and forehead) and activities (head motions while standing and walking) differed in the previous studies, compared with our study on IMU earbud sensor use during workout activities [24,37,38,39]. Although the IMU earbud sensor was less stable than a sensor attached to the forehead, and the workout activities (single-leg squats, lunges, and standing hip abduction) were less-stable motions than standing and walking, the earbud IMU sensor exhibited similar moderate-to-very-strong validity for measuring head angles. Therefore, wireless earbuds may be useful for providing appropriate exercise movements based on the head angle, particularly when used with future machine-learning algorithms.

### 4.2. Pairwise Correlations between Head Angle and Angles of Other Joints

We found that head movement was similar to trunk movement in only the frontal plane, based on data obtained from the 3D motion analysis system during the single-leg squat, reverse lunge, and standing hip abduction exercises. Consistent with these results, the frontal head angle measured by the earbud-type IMU sensor showed a similar pattern to the frontal trunk angle measured with the 3D motion analysis system, based on the fair correlation between head and trunk angles. There are two plausible reasons for these observations. First, the trunk is closer to the head than to the hips and knees, resulting in a fair correlation between the trunk and head angles. The second reason may be explained by vestibular function in healthy people. Our observations are consistent with previous findings that healthy people have a greater head–trunk correlation than people with unilateral vestibular hypofunction while walking on a treadmill because the vestibular system controls movements of the head and trunk together through the vestibulo-cervical reflex and vestibulospinal reflex, respectively [40]. In people with vestibular system impairment, the head and trunk tend to move independently because they are controlled by other signals, such as visual signals and somatosensory information, respectively [41]. All participants in the present study were healthy without vestibular function impairment; they showed similar patterns of movement between the head and trunk.

An earbud IMU sensor could be used for measuring the trunk motion indirectly in clinical and sports settings. Specifically, earbud IMU sensors can be used to prevent knee injury by assessing frontal trunk motion during functional performance tests, such as single-leg squats and forward step-down tests, because frontal trunk motions can directly influence the knee moment in the frontal plane; people with knee pain reportedly exhibited greater ipsilateral trunk lean than did healthy controls [33,42]. An earbud IMU sensor can also provide real-time music-based feedback for effective training to reduce excessive trunk lateral lean while contracting weak hip muscles for knee rehabilitation because excessive trunk lateral lean means poorer hip abductor and external rotator function during workout activities (single-leg squats, lunges, and standing hip abduction) [33]; for example, music played through the wireless earbud during a workout could be distorted if trunk lateral lean occurs during single-leg squats and lunges. However, in order to use earbud IMU sensors for detecting the excessive trunk lateral leaning during workout activities based on the correlation of trunk motion with head motion in the current study, an additional study would be needed. In the future study, classification model should be developed based on the frontal head angle measured by the earbud-type IMU sensor to discriminate between excessive and non- excessive trunk lateral leaning during single-leg squats, lunges, and standing hip abduction, in people with and without lower back and lower limb pain using supervised machine learning methods such as logistic regressions, decision trees, random forests, support vector machines, and k-nearest neighbours. This future classification study also could suggest the cut-off values of frontal head motion to discriminate between excessive and non-excessive trunk lateral leaning.

An earbud IMU sensor may also be useful for vestibular rehabilitation using visual and auditory feedback. A previous study provided visual feedback on the anteroposterior and mediolateral displacement of the trunk on a TV screen using two webcams and showed that a 4-week visual feedback program significantly improved balance in older people with vestibular loss or sensory impairment by increasing their awareness of body motion while walking. In another auditory feedback study, the volumes of four loudspeakers increased linearly, from 60 to 95 dB, towards the swaying side depending on the trunk angle velocity, improving postural control in patients with vestibular dysfunction [43]. In an example of harnessing this feedback methodology in the current earbud IMU sensors, the virtual head–trunk image can tilt in a visual display for visual feedback or the volume of music heard through the earbud can change to provide auditory feedback depending on the extent of mediolateral head–trunk sway while walking; this could improve the balance of people with vestibular dysfunction by maintaining upright head–trunk motion.

This study found poor pairwise correlations between frontal head angle (measured by the earbud-type IMU sensor) and the hip and knee angles (measured by the 3D motion analysis system). To our knowledge, there have been no studies regarding direct correlations of movement between the head and other joints. However, some studies reported the pattern of trunk motion, compared with lower limb motion, in terms of compensatory mechanisms [44,45]. In people with patellofemoral pain syndrome, the trunk moves more laterally towards the stance limb, while the pelvis and hips move in the opposite direction (i.e., contralateral pelvic drop and hip adduction), compared with movement in healthy people [46]. Consistent with these results, in people with knee osteoarthritis, the trunk leans laterally to gain mechanical benefit for the knee via compensatory mechanisms [47,48]. However, it would have been unnecessary for the healthy participants in the present study to use compensatory trunk leaning, regardless of lower limb movement, to reduce the moment of the painful hip and knee during workout activities; this presumably led to poor correlations between the head–trunk complex and hips/knees.

There was no significant correlation between head and trunk motions in the sagittal plane during workout activities in this study, meaning that the earbud IMU sensor cannot inform excessive trunk flexion or extension during squats, single-leg squats, lunges, and standing hip abduction. Participants had to gaze at the front during performing workout activities in this study, so the head was maintained without excessive head flexion or extension even though the trunk was flexed or extended during performing squat and lunge. This may be the reason why no significant correlation was found between head and trunk motions in the sagittal plane in the current study. This study found correlations only in the sagittal and frontal planes because a systematic review found low validity for the rotation components of the trunk and lower limb joints [49], and trunk lateral leaning in the frontal plane has commonly been considered as the target movement that needs to be detected during rehabilitation exercise in previous studies rather than sagittal and transverse planes in people with pain in the lower back, knee, and ankle [33,42,50,51]. However, another study reported that runners who performed single-leg squats poorly had excessive trunk rotation in the transverse plane towards the stance limb [52], indicating the need to measure motion with three degrees of freedom using wearable sensors in the future study.

### 4.3. Limitations

First, the participants in this study were young active people who exercised regularly and had no musculoskeletal pain. Therefore, the results cannot be generalised to sedentary or inactive people, older adults, and people with musculoskeletal pain. Second, only a gyroscope and an accelerometer were used to maximise battery life, with the aim of facilitating the commercial development of the wireless earbud-type IMU sensor. A previous study suggested that using a gyroscope, accelerometer, and magnetometer together can overcome the limitations of each sensor type and improve measurement accuracy, although sensor fusion could not completely compensate for all sensor limitations [53]. Further validation studies are needed to support applications of 3D body motion tracking using nine-axis IMU sensors that include a gyroscope, accelerometer, and magnetometer in people with knee pain or sedentary lifestyles, or professional athletes.

## 5. Conclusions

The wireless earbud-type IMU sensor showed moderate-to-very-strong concurrent validity in head angle measurement with 3D motion analysis results in sagittal and frontal planes during the squat, single-leg squat, lunge, and standing hip abduction exercises. The earbud-type IMU sensor can be used for indirect measurement of frontal trunk motion; it showed a fair correlation between the head and trunk in the frontal plane during workout activities. These findings suggest that a wireless earbud incorporating a single IMU sensor could be useful for measuring the head motion directly or frontal trunk motion indirectly in workout and sport settings.

## Figures and Tables

**Figure 1 sensors-22-00597-f001:**
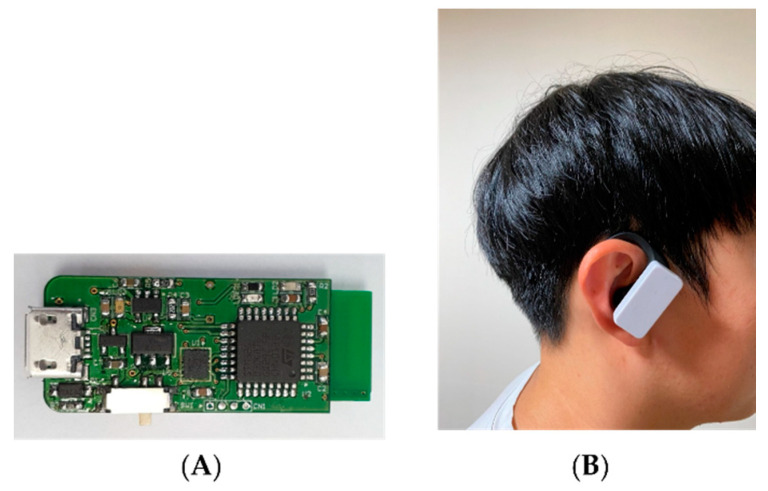
Earbud-type IMU sensor. (**A**) IMU sensor. (**B**) IMU sensor attached to the ear.

**Figure 2 sensors-22-00597-f002:**
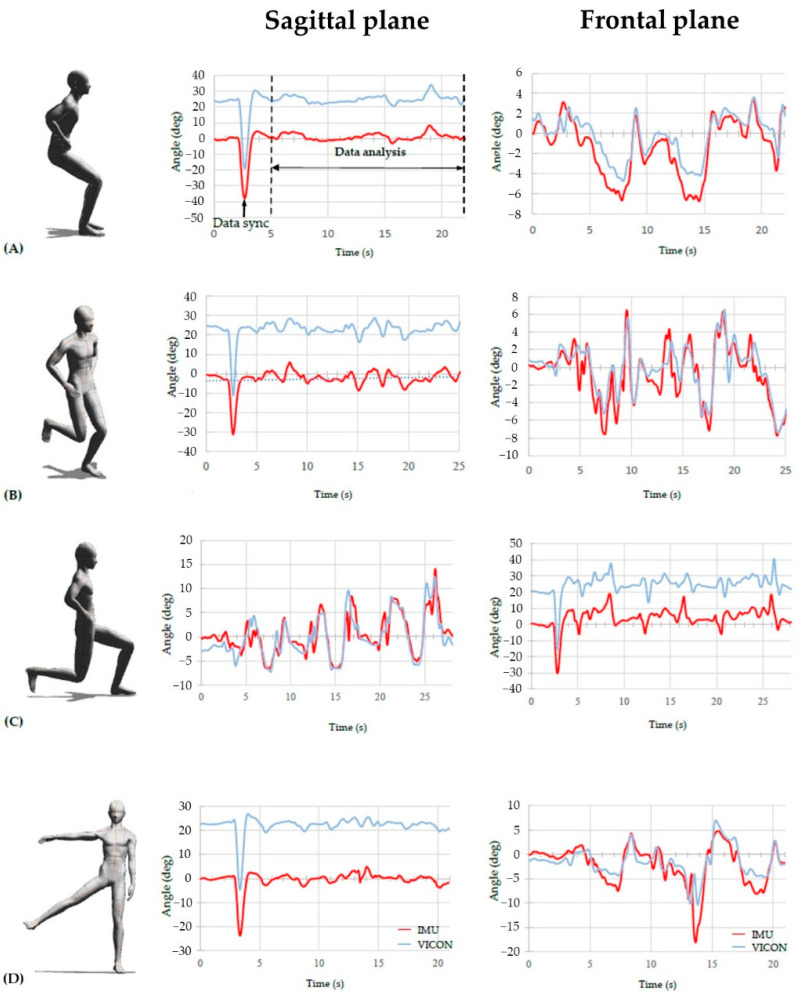
Comparisons of head angle between earbud-type IMU sensor and the 3D motion analysis system during each workout in sagittal plane and frontal plane. The two datasets were synchronised with each peak of rapid neck flexion (data sync.): (**A**) squat, (**B**) single-leg squat, (**C**) reverse lunge, and (**D**) standing hip abduction.

**Table 1 sensors-22-00597-t001:** Demographic characteristics of the participants.

Variables	Mean ± Standard Deviation
Male/female, *n*	9/11
Age, years	21.1 ± 3.1
Weight, kg	63.7 ± 11.3
Height, cm	169.3 ± 7.8
Body mass index, kg/m^2^	22.1 ± 2.6
Exercise frequency per week, times/week	3.6 ± 1.7
Daily exercise duration, min	69.5 ± 33.3

**Table 2 sensors-22-00597-t002:** Concurrent validity of head angle determined with the earbud-type IMU sensor, compared with the 3D motion system.

	Sagittal Plane		Frontal Plane	
	Spearman’s ρ	95% CI	Spearman’s ρ	95% CI
Squat	0.906 *	0.861, 0.951	0.760 *	0.695, 0.825
Single-leg squat	0.897 *	0.879, 0.915	0.753 *	0.706, 0.799
Reverse lunge	0.624 *	0.56, 0.689	0.784 *	0.74, 0.828
Standing hip abduction	0.912 *	0.887, 0.938	0.766 *	0.691, 0.841

* *p*-values were significant (*p* < 0.05).

## Data Availability

The data presented in this study are available on request from the corresponding author due to containing data from human volunteers.

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
