# Peer review of "The Validity of Wireless Earbud-Type Wearable Sensors for Head Angle Estimation and the Relationships of Head with Trunk, Pelvis, Hip, and Knee during Workouts"

_sensors, 2022, doi:10.3390/s22020597_

Round 1

Reviewer 1 Report

The authors have presented an experimental study on the correlation of head angles with other anatomical articulations angles to draw conclusions about the usability of these correlations to give users feedback about the activity that they are performing.

The paper shows an incrementally new idea and tests it on a specific case of indoor exercises. In the introduction, the authors highlight the limitations of smartwatches in the determination of incorrect trunk angles, whereas head angles can be more directly related to such motion.

The main limitation I see is the fact that the correlation of trunk motion with head motion is not further analyzed to draw conclusions about incorrect movements. One of the main mistakes one can do during squat, for example, is bending the spinal cord, thus increasing dramatically the load on the lumbar spine discs. In the study, only a general correlation of sagittal plane and frontal plane angles is reported, but the motion of the trunk cannot be limited to a 2 DoFs motion.

The authors should also provide more details about the calculation of the angles and the effectiveness of the calibration and filtering that they adopt. For the calibration, what happens if the sensor is not perfectly aligned, or if it moves during the exercise (do the authors report any of these occurrences?)? For filtering, did the authors check the goodness of the reconstructed g vector? More details should also be reported for the calculation of the trunk angles.

The discussion introduces the case of vestibular hypofunction, but they d not discuss thoroughly how the obtained results could be used for providing the user with feedback.

Author Response

1) One of the main mistakes one can do during squat, for example, is bending the spinal cord, thus increasing dramatically the load on the lumbar spine discs. In the study, only a general correlation of sagittal plane and frontal plane angles is reported, but the motion of the trunk cannot be limited to a 2 DoFs motion.

-Line 375-384: We agree that 3D measurement should be needed during performing workout activities, so we added the importance of trunk motion measurement in 3D planes. We added this in limitation and necessary of further validity study using three degrees-of-freedom motion.

2) The authors should also provide more details about the calculation of the angles and the effectiveness of the calibration and filtering that they adopt. For the calibration, what happens if the sensor is not perfectly aligned, or if it moves during the exercise (do the authors report any of these occurrences?)? For filtering, did the authors check the goodness of the reconstructed g vector? More details should also be reported for the calculation of the trunk angles.

- Line 133-152: We added the details about the calibration and filtering.

3) The discussion introduces the case of vestibular hypofunction, but they d not discuss thoroughly how the obtained results could be used for providing the user with feedback.

-We explained how to provide auditory or visual feedback about head-trunk motions for both healthy people (Line 331-337) and people with vestibular hypofunction (Line 337-350) based on the previous study.

Reviewer 2 Report

The paper presents a study to investigate the validity of the use of an inertial measurement unit for estimating the head angle of an individual when performing a set of different physical exercises. I think the topic is very interesting and suited for the Applications of Body Worn Sensors and Wearables special issue. 

On the other handI think that some topics must be better presented prior the paper can be considered for publication. I will enumerate them below:

1) The special issue is focused on wearables to facilitate management of chronic conditions or preventive care. On the other hand, the study was made only with very fit and young individuals without any actual or previous disease. The reasons for choosing these individuals must be clarified. Moreover, at the introduction, no justificative was made about the usefullness of the results for preventing or detecting any health condition. This way I suggest a deep reformulation of the introduction to better fit the paper in the special issue.

2) The paper is very focused on the methods for collecting the data and discussing the results. The signal processing used to transform the raw data in useful information is only mentioned without developmentdiscussion or justificative. Important examples of this are the formulas to calculate the saggital and frontal angles and the filters used.

3) The correlation thresholds chosen by the researchers can change all the conclusions for the paper. Why a Spearman rho correlation coefficient of 0.8 may be considered very strong? The assumptions of strong or weak correlation must be made by an external reference and not by the researchers.

4) The proposed technique is not compared with any other in the literature. Previous studies are briefly cited in section 4.1. A better literature review must be performed, and an external benchmark must be provided.

Author Response

1) The special issue is focused on wearables to facilitate management of chronic conditions or preventive care. On the other hand, the study was made only with very fit and young individuals without any actual or previous disease. The reasons for choosing these individuals must be clarified. Moreover, at the introduction, no justificative was made about the usefulness of the results for preventing or detecting any health condition. This way I suggest a deep reformulation of the introduction to better fit the paper in the special issue.

- We reformatted the introduction to fit the special issue.

1) In line 37-46, we added the usefulness of wearable sensor for chronic musculoskeletal pain.

2) In line 63-75, we explained the characteristics of trunk and head motion in chronic ankle and knee pain, Parkinson's disease, multiple sclerosis and vestibular impairment.

3) In line 79-89, we added the role of wearable sensors for injury prevention; specifically, wearable sensor can be used for detecting future musculoskeletal pain in present healthy active adults, so this study investigated the validity of earbud-type IMU sensor prior to future study.

2) The paper is very focused on the methods for collecting the data and discussing the results. The signal processing used to transform the raw data in useful information is only mentioned without development, discussion or justificative. Important examples of this are the formulas to calculate the sagittal and frontal angles and the filters used.

- Line 133-152: We added the signal processing used to transform the raw data using formula.

3) The correlation thresholds chosen by the researchers can change all the conclusions for the paper. Why a Spearman rho correlation coefficient of 0.8 may be considered very strong? The assumptions of strong or weak correlation must be made by an external reference and not by the researchers.

-Line 233: We added the reference about the criteria of Spearman rho.

4) The proposed technique is not compared with any other in the literature. Previous studies are briefly cited in section 4.1. A better literature review must be performed, and an external benchmark must be provided.

- Line 288-303: We added the discussion with external benchmark based on additional previous studies.

Round 2

Reviewer 1 Report

The authors only addressed the concerns raised in the first review round, but the answers acknowledge the issues instead of solving them. The design has the same lack that I highlighted, the calibration was not assessed against possible errors, filtering reports more mathematical details, but no details on the goodness of the obtained result.

Author Response

1) The authors only addressed the concerns raised in the first review round, but the answers acknowledge the issues instead of solving them.

  • We appreciate your comments. We strived to solve the concern raised during the first and second review periods and answered like below. If the corrected paper is not proper based on your point of view, please let me know which parts should be changed.
  • You have raised an important point that the correlation of trunk motion with head motion is not further analyzed to draw conclusions about incorrect movements in the first review round. As your comments, our data cannot be further analyzed to draw conclusions about detecting incorrect trunk motion using Ear IMU sensor, because subjects were healthy who incorrect motion might not be observed excessively (Line 373-376). Previous studies demonstrated that healthy people showed less trunk lateral leaning during single-leg squat and single-leg drop vertical jump than participants with back and knee pain (Line 368-372).
  • Thus, to avoid hasty generalization, we corrected or deleted the sentences that wearable sensor is able to provide feedback related to incorrect motion measured by smartwatch and earbud in the abstract (Line 24), introduction (Line 56-58), discussion (Line 276 and 326) and conclusions (Line 415-419), except of Reviewer 2’s response in 1st revision (Line 327-337 and 350-362). And we have clarified the purpose of the current study (Line 86-93) in the introduction.
  • In order to suggest the usability of the correlation of trunk motion with head motion, we added the future study (Line 338-346) as your suggestion which will develop the classification model using supervised machine learning methods based on the head motion data obtained from earbud IMU sensor as likely as below previous literature.1)-3) This future classification study also could suggest the cut-off values of frontal head motion to discriminate between excessive and non-excessive trunk lateral leaning (Line 346-348).
    • Whelan, Darragh, et al. "Leveraging IMU data for accurate exercise performance classification and musculoskeletal injury risk screening." 2016 38th Annual International Conference of the IEEE Engineering in Medicine and Biology Society (EMBC). IEEE, 2016.
    • Chen, Yin-Jun, and Yen-Chu Hung. "Using real-time acceleration data for exercise movement training with a decision tree approach." Expert Systems with Applications 37.12 (2010): 7552-7556.
    • Caserman, Polona, Clemens Krug, and Stefan Göbel. "Recognizing Full-Body Exercise Execution Errors Using the Teslasuit." Sensors 21.24 (2021): 8389.

2) Reviewer suggested a useful example in the first revision that trunk flexion as incorrect motion during a squat can increase the load on the lumbar spine, resulting in back pain. And reviewer also suggested that trunk motion is cannot be limited to a 2 DoFs motion.

  • We add the below explanation in order not to exaggerate the results.
    • Because there was no significant correlation between head and trunk motions in the sagittal plane during workout activities in this study, the earbud IMU sensor cannot inform excessive trunk motion in the sagittal plane during squat or single leg squat (Line 378-381)
  • We also add the explanation of why trunk motion was not similar to head motion in the sagittal plane (Line 381-385). Additionally, we explain the necessity of measurement in the transverse plane with the previous reference (Line 385-390), although some studies have commonly been measured in the frontal plane in people with pain in the lower back, knee, and ankle (Line 391-394).

3) The design has the same lack that I highlighted, the calibration was not assessed against possible errors, filtering reports more mathematical details, but no details on the goodness of the

  • We add the explanation about the calibration which was not assessed against possible errors (Line 152-155).
    • During 1 s calibration period, a subject was told to remain stationary, 100 data samples collected and assessed to confirm calibration. Only when (Maxϕ-Minϕ) and (Maxθ-Minθ) is less than 0.5°, it was accepted as a successful calibration. If not, a test started over from the calibration period.
  • We have clarified about the filtering (Line 146-149).
    • To properly calculate roll and pitch angle from accelerometer, magnitude of accelerometer output should be near 1G, which means only gravitational acceleration exist. After low-pass filtering, magnitude of accelerometer output converged to 1G within 5% difference.

Reviewer 2 Report

I have carefully read the answers for my previous inquiries and they are sufficient. The second version of the manuscript has improved its quality and, this way I have no objections about paper publication.

Author Response

We appreciate your comments and suggestions concerning our manuscript.